# Identification of Causal Agent Inciting Powdery Mildew on Common Bean and Screening of Resistance Cultivars

**DOI:** 10.3390/plants11070874

**Published:** 2022-03-25

**Authors:** Dong Deng, Suli Sun, Wenqi Wu, Canxing Duan, Zhaoli Wang, Shilong Zhang, Zhendong Zhu

**Affiliations:** 1Institute of Crop Sciences, Chinese Academy of Agricultural Sciences, Beijing 100081, China; 82101219111@caas.cn (D.D.); wuwenqi@caas.cn (W.W.); duancanxing@caas.cn (C.D.); 2Coarse Cereal Unit, Bijie Academy of Agricultural Sciences, Bijie 551700, China; nksdezb@126.com (Z.W.); bjrice@163.com (S.Z.)

**Keywords:** *Phaseolus vulgaris*, powdery mildew, *Erysiphe vignae*, phylogenetic analysis, disease resistance

## Abstract

Powdery mildew is one of the severe diseases on common bean in Southwestern China, but the identity of the pathogen inciting this disease is unclear. The objective of this study was to identify the causal agent of common bean powdery mildew and to screen resistant cultivars. The pathogen was identified through morphological identification, molecular phylogenetic analysis, and pathogenicity tests. Resistance of common bean cultivars was evaluated by artificial inoculation at the seedling stage. The common bean powdery mildew isolate CBPM1 was obtained after pathogen isolation and purification. Morphological identification confirmed that the isolate CBPM1 belonged to the *Oidium* subgenus *Pseudoidium* and germinated *Pseudoidium*-type germ tubes. Molecular phylogenetic analysis showed that the isolate CBPM1 and *Erysiphe vignae* isolates from different hosts were clustered into a distinct group. The pathogenicity and host range tests revealed that the isolate CBPM1 was strongly pathogenic to common bean, multiflora bean, lablab bean, cowpea, and mung bean, but not to soybean, adzuki bean, pea, faba bean, chickpea, lentil, pumpkin, and cucumber. In addition, 54 common bean cultivars were identified for resistance to powdery mildew, and 15 were resistant or segregant. Based on the morphological, molecular and pathogenic characteristics, the causal agent of common bean powdery mildew was identified as *E*. *vignae*. This is the first time *E*. *vignae* has been confirmed on common bean. Cultivars with different resistance levels were screened, and these cultivars could be used for disease control or the breeding of new resistant cultivars.

## 1. Introduction

Common bean (*Phaseolus vulgaris* L.), commonly known as kidney bean in China, is the largest edible legume crop for direct human consumption [1]. Common bean contributes to human health and nutritional security as it is well-endowed with protein, minerals, dietary fiber, vitamins, and other nutrients, and it also can improve soil fertility by N assimilation due to symbiosis [1,2,3,4].

China is a major producer of common bean, with the production of 1,294,370 tons in 2020, ranking fifth among the 106 countries growing common bean in the world [5,6]. The production of common bean is restricted by a variety of biotic and abiotic factors, among which diseases are the main factors affecting the yield and quality of common bean in China, such as common bacterial blight, Fusarium wilt, and angular leaf spot [7,8,9]. In addition, during our recent disease investigation in 2020, we found that powdery mildew is also one of the important diseases of common bean in Southwestern China, especially in Guizhou Province. So far, powdery mildew has been documented in snap bean in Hebei, Jilin and Shanxi provinces, but has not been reported in common bean in China [10]. Nevertheless, powdery mildew has always been one of the major yield limiting factors in some common bean production regions in the world, such as Brazil, Mexico, Spain, and the United States, and can cause up to 69% of yield loss [11,12].

The agent of powdery mildew on common bean is generally considered to be *Erysiphe polygoni* [10,13,14], but with the application of molecular phylogenetic analysis, the taxonomic status of this pathogen has been redefined. Almeida et al. [15] found that *Erysiphe* sp. isolate EB2004 (Genbank accession No. AY739109), a common bean powdery mildew isolate in Brazil, had the most closely genetic relationship with the soybean powdery mildew *E. diffusa* by analyzing the internal transcribed spacer (ITS) sequences of the nuclear ribosomal DNA (nrDNA) [16]. Taking *Erysiphe* sp. isolate EB2004 as a reference, Felix-Gastelum et al. [12], Trabanco et al. [17], and Leitão et al. [18] identified the common bean powdery mildew pathogen as *E. diffusa* in Mexico, Spain, and Portugal, respectively. Campa and Ferreira [19] re-sequenced the ITS region of the isolate from Trabanco et al. [17] and found that the isolate (KU320678) had the greatest similarity with *E. polygoni*, and considered the isolate to be *E. polygoni*. Interestingly, Kelly et al. [20] recently found that the ITS sequence of a new powdery mildew species, *E. vignae*, identified on mung bean, was identical to that of a Spanish bean isolate PMbean-1 and the reference isolate EB2004. A phylogenetic tree was constructed based on the ITS sequences to cluster the two common bean powdery mildew isolates and the *E. diffusa* isolates from Australian and American soybeans into two distinct phylogenetic groups. Therefore, Kelly et al. [20] renamed the two *E. diffusa* isolates in Brazil and Spain as *E. vignae* ex *Phaseolus vulgaris*.

Powdery mildew is the most common and important disease in agricultural production. Although some biological and chemical measures can effectively control the occurrence of powdery mildew, the use of resistant cultivars is the most effective, economical, and environmentally friendly method to control this disease [21]. At present, common bean powdery mildew occurs commonly in the Yunnan–Guizhou Plateau, and is especially severe in Bijie City, Guizhou Province, but the pathogen has not been clarified and the identification of resistance cultivars has not been carried out. Thus, the objective of this study was to identify pathogen species inciting powdery mildew of common bean through morphological identification, molecular phylogenetic analysis, and pathogenicity and host range tests, and to screen for resistance in common bean cultivars.

## 2. Results

### 2.1. Disease Symptoms and Morphological Identification

The infected leaves initially showed slightly darkened spots on the upper surface, which then developed into white mildew blotches (Figure 1A). The blotches expanded and merged to form a white mildew layer covering the whole leaf surface (Figure 1B,C). Severely infected leaves turned yellow, died, and fell off (Figure 1C). As the disease progressed, the pods and stems became infected, and eventually the diseased tissues were covered with white mildew (Figure 1B).

The conidiophores of the obtained isolate CBPM1 were cylindrical and erect with 44.2–76.0 × 8.0–10.3 μm in size, composed of one foot cell and 1–2 short cells, and produced individual conidia. The foot cells were cylindrical, straight or slightly curved, with 22.3–30.0 × 8.0–10.1 μm in size (Figure 2A). The conidia were elliptic to ovate, and 25.4–35.8 × 13.5–19.8 μm in size (Figure 2B). The mycelial appressoria were lobed (Figure 2C). All these characteristics confirmed that the isolate CBPM1 belonged to the *Oidium* subgenus *Pseudoidium*.

After 48 h incubation of conidia on glass slides and the inner surface of petri dish lids, the lengths of germ tubes produced by 90% germinated conidia were shorter than or equivalent to that of conidia. The germ tubes grew from the terminal or near terminal of the conidia, and the apices of germ tubes were lobed or multi-lobed (Figure 2D,F). This result indicated that the conidia of isolate CBPM1 germinated the *Pseudoidium*-type germ tubes.

### 2.2. Molecular Identification and Phylogenetic Analysis

The ITS region of the isolate CBPM1 was amplified with the primer pair PMITS1/PMITS2, and the 756 bp target sequence was obtained (MW579545). The ITS sequences of CBPM1 were aligned and analyzed on NCBI and were found to be 100% similar to 14 ITS sequences in Genbank. These sequences belonged to two powdery mildew species, *E. diffusa* and *E. vignae*, respectively. *E. diffusa* included five common bean powdery mildew isolates from Brazil, Mexico, and Spain (AY739109, HQ444195, HQ4441957, HQ444198 HQ444195, KU320678), one cowpea powdery mildew isolate from Brazil (KY515231), and one powdery mildew isolate with unknown geographic and host origin (MT878222). *E*. *vignae* included five mung bean powdery mildew isolates from Australia (MT628282, MT628285, MT628286, MW293894, MW293895), one black gram powdery mildew isolate (MT628284), and one soybean powdery mildew isolate from China (MG171170). The corresponding powdery mildew ITS sequences were obtained from the GenBank to construct a phylogenetic tree. It was found that *E. diffusa* and *E. vignae* from different host plants were clustered in a large phylogenetic group, the isolate CBPM1 was completely consistent with six *E. vignae* isolates and eight renamed *E. vignae* isolates, which used to be considered as *E. diffusa* or unidentified [20], clustered in one subgroup, while other *E. diffusa* isolates clustered in another subgroup (Figure 3). The confidence of the results was 100%.

The 5′-end of the nrDNA large subunit (LSU) gene sequences alignment analysis showed that the sequence of CBPM1 was 100% identical to the two *E. vignae* isolates of mung bean from Australia in Genbank. A phylogenetic tree was constructed based on LSU sequences. The isolate CBPM1, two *E. vignae* isolates of mung bean (MT628017, MT628018) and one *E. diffusa* isolate of soybean from Australia (MT628019) clustered in one phylogenetic group, but CBPM1 and two *E. vignae* isolates clustered in one subgroup, *E. diffusa* isolates were in a single subgroup (Figure 4). The results also supported the clustering results of ITS sequences.

### 2.3. Pathogenicity and Host Range Tests

The four common bean cultivars were infected by shaking conidia from heavily infected plants. After 15 days, the cultivars ‘Yingguohong’ and ‘Pinjinyun 5’ were covered with a white mildew layer, and the infection types (IT) were four; cultivars ‘F3370’ and ‘F5033’ have showed necrotic reaction and moderate mycelia, and the infection types were two. The young leaves, stems, and pods were also naturally infected by powdery mildew with the plant growth. These results indicated that the isolate CBPM1 was pathogenic to common bean.

In the host range tests on 12 other crops, the isolate CBPM1 showed no pathogenicity to seven crops including soybean, chickpea, lentil, faba bean, pea, pumpkin, and cucumber, with no symptoms observed on the inoculated plants. Conversely, the isolate CBPM1 was highly pathogenic to mung bean, cowpea, multiflora bean, and lablab bean (Table 2). There were four infection types of four mung bean cultivars, which showed typical symptoms on leaves covered with abundant mycelial and conidia. Three cowpea cultivars were covered by abundant or moderate mycelial and conidia with IT4 or IT3, while the plants of cowpea cultivar ‘Jijiang 3’ showed a necrotic reaction and little mycelia with IT1. Most plants of two lablab bean cultivars were IT4 or IT3, whereas the rest were IT0. Conversely, a small number of plants of multiflora bean cultivar ‘18E07’ were IT4 or IT3, but most were IT0. The inoculated plants of four adzuki bean cultivars showed visible symptoms, but there were sparsely conidium-producing mycelium on the stems of cultivar ‘Baihong 12’ and brown lesions on the stems of cultivars ‘Baohong 201206-5’ and ‘Liaohong 12814’.

### 2.4. Resistance Evaluation of Common Bean Cultivars

The resistance of 54 common bean cultivars to powdery mildew isolate CBPM1 was identified, and 15 cultivars showed resistance or segregation, accounting for 27.8% of all identifications (Table 3). Cultivars ‘ZYD19-01’, ‘LiBY-4’, ‘LiBY-6’, and ‘LiBY-9’ were segregant and produced two types of immune and highly susceptible seedlings with IT0 and IT4, respectively. There was one infection type of seven cultivars, including ‘Long 15-1909’, ‘Long 17-4167’, ‘Longyundou 4’, ‘Longyundou 10’, ‘Longyundou 18’, ‘Keyun 3’, and ‘LiBY-5’, and they exhibited high resistance. The ‘Long 16-3263’ was also a segregant cultivar with IT1 and IT3 plants, which were highly resistant and susceptible, respectively. The infection types of four cultivars ‘F5033’, ‘F3370’, ‘Pinyun 2’, and ‘Longyundou 14’ were two and they showed resistance to CBPM1. Among the remaining 39 cultivars, the infection types of eight and 31 cultivars were three and four, respectively, showing that they were susceptible and highly susceptible.

## 3. Discussion

Bijie city, located at high altitude in the northwest of Guizhou Province, is a traditional planting area for common bean. In Weining County, Bijie City, the annual planting area of common bean alone is about 46,700 hm^2^ per year [36,37]. In Guizhou Province, the majority of common bean is interplanted with maize. Due to warm climates, high humidity, and shade environment, it is suitable for the occurrence of powdery mildew [37]. Intercropping might reduce powdery mildew severity [30], but made it difficult to control powdery mildew using conventional disease control measures. Therefore, deploying resistant cultivars is the most effective, economical, and environmentally safe method to control powdery mildew in maize–common interplanting systems. The pathogen of common bean powdery mildew in Bijie City has not been identified to date, which greatly restricts the breeding and utilization of resistant cultivars.

The identification of *Erysiphe* spp. was traditionally based on several main morphological characteristics, including appendage type, conidiophore type, foot cell type, ascus number of per cleistothecium, conidia germination mode, and the presence of fibrous bodies in the conidia [38,39]. However, the cleistothecia of some powdery mildew is hard to produce, such as *E. polygoni* and *E. diffusa*, which greatly reduces the accuracy of morphological identification [39,40]. With the development of molecular biology, molecular characteristic analysis has become an important auxiliary method for pathogen identification. Molecular phylogenetic analysis based on ITS gene has been widely used for the classification and identification of powdery mildew [29,39,40]. In this study, we found the size of conidiophores, foot cells and conidia, and mycelial appressoria of isolate CBPM1 were similar to these of *Oidium* subgenus *Pseudoidium*. Further, most of the germ tubes produced by conidia of isolate CBPM1 were shorter than the length of conidia or as long as the conidia, and the germ tubes were lobed or multi-lobed at the apices after 48 h incubation. These morphological characteristics were the same as those of *E. vignae* [20]. Molecular identification showed that the ITS and LSU sequences of the isolate CBPM1 were consistent with all *E. vignae* isolates currently available in Genbank. Molecular phylogenetic analysis based on ITS and LSU sequences classified the isolates of *E. vignae* and *E*. *diffusa* into two distinct taxa (Figure 3). Therefore, based on the morphological characteristics, conidial germination pattern, and molecular phylogenetic analysis, the isolate CBPM1 was identified as *E. vignae*.

Pathogenicity and host range tests were essential to identify the species and host specialization of plant pathogens. Kelly et al. [20] reported that *E. vignae* was pathogenic to mung bean, provoked a hypersensitive response in cowpea, and no response in soybean. Thirteen crops were tested in this study. The isolate CBPM1 was shown to be highly pathogenic to common bean, cowpea, mung bean, multiflora bean, and lablab bean, but not to the other eight crops. There was one infection type of a cowpea cultivar “Jijin 3” tested; the mixed infection types of multiflora bean and lablab bean cultivars were also detected. These results were consistent with Kelly et al. [20] and confirmed that *E. vignae* was a pathogen of common bean, cowpea, multiflora bean, and lablab bean as well.

*E. diffusa* has been reported to cause powdery mildew on more than 50 species of legumes, and has been widely identified as the pathogen of soybean powdery mildew [15,16,26,41,42]. At present, *E. diffusa* was reported to cause powdery mildew on three legumes in China, including herba thermepsidis (*Thermopsis lanceolata*) in Inner Mongolia [43], soybean in Jining [44], and wisteria in Yunnan Province [27]. In addition, the ITS sequences of a soybean powdery mildew isolate HD-3 (KC832863) from Guizhou Province, China and a Chinese qianjinbo (*Flemingia prostrata*) powdery mildew isolate HMJAU-PM91877 (MT329758) from Sichuan Province, China were available in Genbank and, based on their ITS sequences, both isolates were identified as *E*. *diffusa* as well. However, Kelly et al. [20] found the Chinese soybean isolate HD-3 (KC832863) clustered in the same phylogenetic group as *E. vignae* in phylogenetic analysis, and renamed HD-3 as *E. vignae* ex *Glycine max*. In this study, ITS sequence phylogenetic analysis also clustered isolate HD-3 (MG171170) in the same phylogenetic group with isolate CBPM1, while Chinese qianjinbo isolate HMJAU-PM91877, soybean isolate HMJAU01514, and wisteria isolate HMJAU02177 and other hosts *E. diffusa* isolates formed another phylogenetic group (Figure 3). Kelly et al. [20] also revealed that *E*. *vignae* isolates from mung bean were not pathogenic to soybean cultivar ‘Bunya’, and *E*. *diffusa* isolates from soybean only shown slight symptoms with small sparse colonies and conidia and often causing necrotic reaction in greenhouse cross-inoculation. Our host range tests also revealed that *E*. *vignae* isolate from common bean shown no pathogenicity to two soybean cultivars ‘Williams’ and ‘Huachun 18’, and the result was further confirmed by an inoculation test in ten soybean cultivars, which were highly susceptible to *E*. *diffusa* [45]. Our results and the findings of Kelly et al. suggested that *E*. *vignae* was a pathogen of common bean and soybean in Guizhou Province, but the host specificity of *E*. *vignae* has been evolved in this region [20].

Planting resistant cultivars is the most economical, effective, and green way to control common bean powdery mildew [11]. The screening of common bean accessions for resistance to powdery mildew has been carried out as early as the 1930s, and some resistant accessions had been identified and used [11,13,17,18,21]. In this study, we evaluated the powdery mildew resistance of 54 common bean cultivars, and found that seven cultivars were highly resistant and four cultivars were resistant. In addition, five cultivars showed mixed infection types, which suggest resistance heterozygosity. These results indicated that there was a high proportion (29.6%) of powdery mildew resistant cultivars in all identified cultivars, which should be directly used for the production or further breeding of new resistant cultivars. It should be noted, however, that although 29.6% resistant cultivars are present in identified cultivars, these resistant cultivars might have a narrow genetic background because most resistant cultivars were bred by two breeding units (Table 3). Hence, in order to ensure the sustainable control of powdery mildew in common bean, it is necessary to explore new resistance resources, and discover the resistance genes and molecular markers to speed up the process of resistance breeding at the same time.

## 4. Materials and Methods

### 4.1. Pathogen Collection and Morphological Identification

Common bean powdery mildew was collected from Bijie City, Guizhou Province (27°17′10″ N, 105°18′15″ E) on 9 October 2020. Fresh diseased leaf samples were delivered to the laboratory for pathogen isolation and purification immediately. The method to purify the pathogen was to shake off the conidia on the diseased leaves and inoculate the seedlings of the common bean cultivar ‘Yingguohong’. After inoculation, the plants were cultivated under a 12 h light period in a light incubator at 20 °C. Conidia producing on a single diseased spot were transferred to ‘Yingguohong’ seedlings for propagation and preservation after 10 days, and the isolate was named CBPM1.

A sterile scalpel was used to scrape the powdery mildew from the fresh lesions, and water was used as a floating carrier. The conidia, conidiophores, and appressoria were observed under a light microscope (Olympus CX31, Tokyo, Japan), and the sizes of the conidia and conidiophore were determined by Mshot MD50 image analysis system.

Fresh conidia produced by the isolate CBPM1 on common bean ‘Yingguohong’ leaves were shaken onto microscope slides and the inner surfaces of plastic petri dish lids, and the slides were then placed in a plastic petri dish lined with a filter paper moistened by sterile water. The petri dish was covered with its lid and sealed with plastic wrap, and finally incubated in an incubator at 22 °C under continuous light [20,38]. After 48 h, the germinations of conidia were observed under a light microscope (Olympus CX31).

### 4.2. Molecular Identification and Phylogenetic Analysis

Mycelia and conidia were scraped from common bean leaves with sterile scalpel. Total genomic DNA of CBPM1 was extracted from mycelium by using the Fungi Genomic DNA Extraction Kit (Solarbio, Beijing, China), according to the manufacturer’s instructions. Partial sequences of the ITS and LSU (including domains D1 and D2) were examined. Primer pairs, PMITS1 (5′-TCGGACTGGCCYAGGGAGA-3′) and PMITS2 (5′-AAGGTTTCTGTAGGTG-3′) for ITS, and T2 (5′-GGGCATGCCTGTTCGAGCGT-3′) and TW14 (5′-GCTATCCTGAGGGAAACTTC-3′) for LSU, were used for polymerase chain reaction (PCR) amplifications [25,46]. PCR reactions were carried out using a Gene Amp 9700 thermocycler (Applied Biosystems, Foster City, CA, USA) in 50 μL reaction mixtures containing: 25 ng of DNA, 2 μL of each primer, 25 μL of 2×Taq PCR Mastermix (TIANGEN, Beijing, China), and 17 μL ddH_2_O. The PCRs were as follows: 94 °C for 10 min; 36 cycles of 30 s at 94 °C, 1 min at 56 °C, and 2 min at 72 °C; finally, 10 min at 72 °C and 4 °C hold. All PCR products were purified and sequenced at Sangon Biotech (Shanghai, China), using the aforementioned primers.

The sequencing results were uploaded to the NCBI database and were compared in the NCBI database (http://www.ncbi.nlm.nih.gov, accessed on 15 March 2021) after splicing and correcting. The related sequences of powdery mildew strains were obtained from the GenBank (Table 1), and phylogenetic trees were constructed using the Neighbor-joining method and the Tamura-Nei distance model in MEGA X with 1000 bootstrap repeats [12,46].

### 4.3. Pathogenicity and Host Range Tests

The common bean cultivars ‘Yingguohong’, ‘Pinjinyun 5’, ‘F3370’, and ‘F5033’ were used for the pathogenicity test. The host range tests were performed on several crops including multiflora bean (*Phaseolus multiflorus*), lablab bean (*Lablab purpureus*), cowpea (*Vigna unguiculata*), mung bean (*Vigna radiata*), adzuki bean (*Vigna angularis*), soybean (*Glycine max*), chickpea (*Cicer arietinum*), lentil (*Lens culinaris*), faba bean (*Vicia faba*), pea (*Pisum sativum*), pumpkin (*Cucurbita moschata*), and cucumber (*Cucumis sativus*). The cultivars and quantities of each plant species were shown in Table 2. Five seeds of each cultivar were planted in a paper cup (1 L) filled with a mix of equal volumes of vermiculite and peat, and the planted cups were placed in the greenhouse at 18–22 °C. Three seedlings were kept after emergence. The common bean cultivar ‘Yingguohong’ was used as the susceptible control for the host range test. The cultivars of common bean, multiflora bean, lablab bean, cowpea, mung bean, adzuki bean, and soybean were inoculated when the primary leaves of the seedlings had fully expanded; chickpea, lentil, faba bean, and pea were inoculated at the fourth or fifth leaf stage; pumpkin and cucumber were inoculated when the main leaves had fully expanded. Inoculation was performed by shaking conidia from heavily infected plants of common bean cultivars ‘Yingguohong’ onto the tested plants [47]. The plants were cultured in a greenhouse at 18–22 °C after inoculation, and the disease infection type was determined after 15 days. The infection types of each inoculated plant were assessed on a 0–4 scale according to Trabanco et al. [17]: IT0, seedlings with no visible symptoms; IT1, seedlings with necrotic reaction and no or little mycelial development; IT2, seedlings with necrotic reaction and moderate mycelial development; IT3, seedlings with moderate mycelial development and little sporulation; IT4, seedlings with abundant mycelial development and profuse sporulation. Seedlings with IT0 were considered as immune (IM), while those with IT1, 2, 3 and 4 were considered as highly resistant (HR), resistant (R), susceptible (S), and highly susceptible (HS), respectively. This test was repeated twice.

### 4.4. Resistance Evaluation of Common Bean Cultivars

A total of 54 common bean cultivars were identified for powdery mildew resistance, and their information is shown in detail in Table 3. Planting, inoculating, and disease evaluation were the same as for the pathogenicity and host range tests. The common bean cultivar ‘Yingguohong’ was also used as the susceptible control. The cultivars and the infection types of 0–2 were identified twice.

## Figures and Tables

**Figure 1 plants-11-00874-f001:**
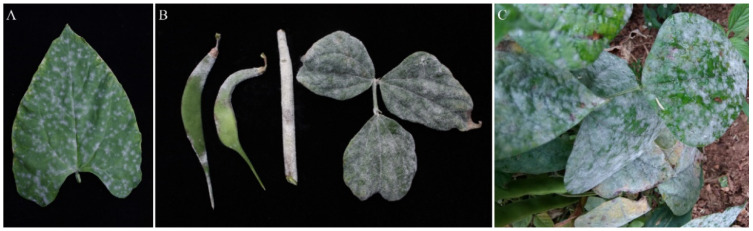
Powdery mildew symptoms on common bean. (**A**) Powdery mildew blotches on a primary leaf by natural infection in glasshouse; (**B**) Symptoms on pods, stem, and leaves developed from pathogenicity test; (**C**) Symptoms on plant observed in field.

**Figure 2 plants-11-00874-f002:**
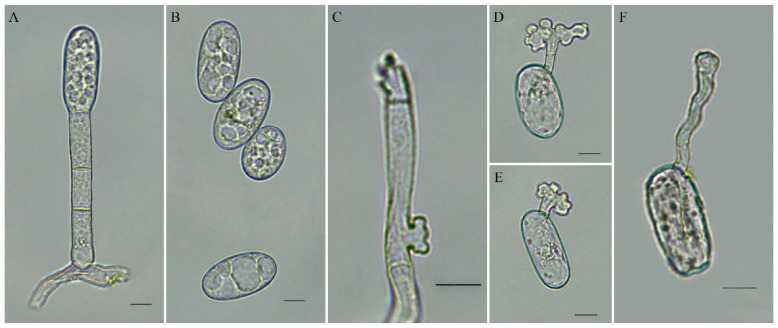
Morphology of *Erysiphe vignae* on common bean. (**A**) Conidiophore with attached conidium; (**B**) Conidia; (**C**) Hyphal appressorium; (**D**,**E**) Conidia with a germ tube germinating on a glass slide; (**F**) Conidium with a germ tube from diseased leaf; Bars = 10 μm.

**Figure 3 plants-11-00874-f003:**
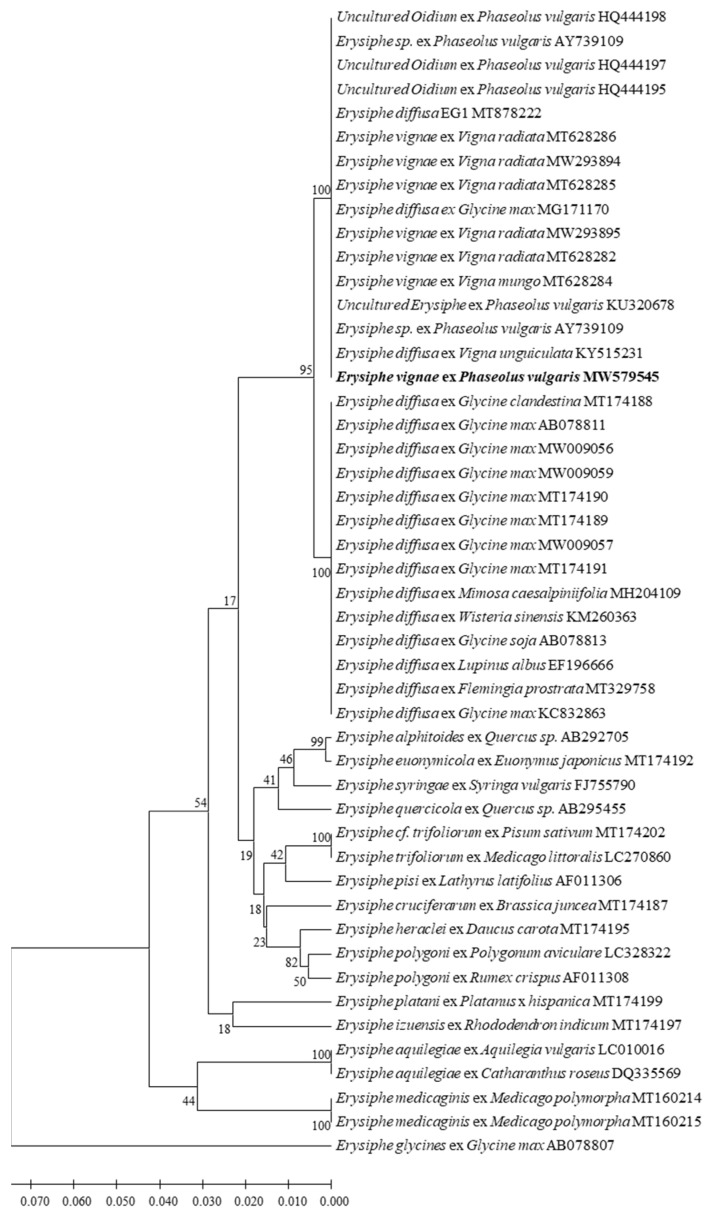
Phylogenetic tree inferred from Neighbor-joining analysis of the ITS sequences in this study and GenBank (Table 1). The tree is rooted to the ITS sequence of *Erysiphe glycines* (AB07807).

**Figure 4 plants-11-00874-f004:**
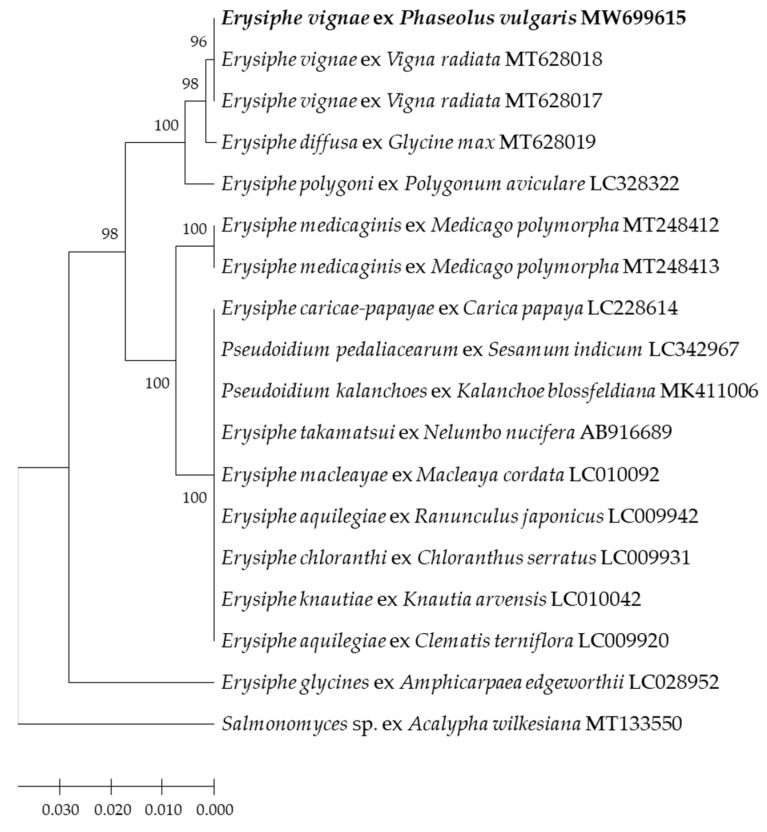
Phylogenetic tree inferred from Neighbor-joining analysis of the LSU sequences in this study and GenBank (Table 1). The tree is rooted to the LSU sequence of *Salmonomyces javanicus* (MT133550).

**Table 1 plants-11-00874-t001:** Powdery mildew species used in the phylogenetic analyses.

Powdery Mildew Species	Host Plant Species	Accession Number of nrDNA		Reference
		ITS	LSU	
*Erysiphe vignae*	*Phaseolus vulgaris*	MW579545	MW699615	In this study
*E. alphitoides*	*Quercus* sp.	AB292705	-	[12]
*E. aquilegiae*	*Catharanthus roseus*	DQ335569	-	[22]
*E. aquilegiae*	*Ranunculus japonicus*	-	LC009942	[23]
*E. aquilegiae*	*Clematis terniflora*	-	LC009920	[23]
*E. aquilegiae*	*Aquilegia vulgaris*	LC010016	-	[23]
*E. caricae-papayae*	*Carica papaya*	-	LC228614	[24]
*E.* cf. *trifoliorum*	*Pisum sativum*	MT174202	-	[25]
*E. chloranthi*	*Chloranthus serratus*	-	LC009931	[23]
*E. cruciferarum*	*Brassica juncea*	MT174187	-	[25]
*E. diffusa*	*Flemingia prostrata*	MT329758	-	Unpublished
*E. diffusa*	*Glycine clandestina*	MT174188	-	[20]
*E. diffusa*	*G. max*	MW009056	MT628019	[20]
*E. diffusa*	*G. max*	MW009057	-	[20]
*E. diffusa*	*G. max*	MW009059	-	[20]
*E. diffusa*	*G. max*	MT174189	-	[25]
*E. diffusa*	*G. max*	MT174190	-	[25]
*E. diffusa*	*G. max*	MT174191	-	[25]
*E. diffusa*	*G. soja*	AB078813	-	[16]
*E. diffusa*	*G. max*	AB078811	-	[16]
*E. diffusa*	*G. max*	KC832863	-	Unpublished
*E. diffusa*	*G. max*	MG171170	-	Unpublished
*E. diffusa*	*Lupinus albus*	EF196666	-	[15]
*E. diffusa*	*Mimosa caesalpiniifolia*	MH204109	-	[26]
*E. diffusa*	*V. unguiculata*	KY515231	-	Unpublished
*E. diffusa*	*Wisteria sinensis*	KM260363	-	[27]
*E. diffusa*	*-*	MT878222	-	Unpublished
*E. euonymicola*	*Euonymus japonicus*	MT174192	-	[25]
*E. glycines*	*G. max*	AB078807	-	[23]
*E. glycines*	*Amphicarpaea edgeworthii*	-	LC028952	[23]
*E. heraclei*	*Daucus carota*	MT174195	-	[25]
*E. izuensis*	*Rhododendron indicum*	MT174197	-	[25]
*E. knautiae*	*Knautia arvensis*	-	LC010042	[23]
*E. macleayae*	*Macleaya cordata*	-	LC010092	[23]
*E. medicaginis*	*M. polymorpha*	MT160214	MT248412	[28]
*E. medicaginis*	*M. polymorpha*	MT160215	MT248413	[28]
*E. pisi*	*Lathyrus latifolius*	AF011306	-	[29]
*E. platani*	*Platanus x hispanica*	MT174199	-	[25]
*E. polygoni*	*Polygonum aviculare*	-	LC328322	[30]
*E. polygoni*	*Rumex crispus*	AF011308	-	[29]
*E. quercicola*	*Quercus* sp.	AB295455	-	[31]
*E. syringae*	*Syringa vulgaris*	FJ755790	-	[32]
*E. takamatsui*	*Nelumbo nucifera*	-	AB916689	[33]
*E. trifoliorum*	*Medicago littoralis*	LC270860	-	[34]
*E. vignae*	*Vigna mungo*	MT628284	-	[20]
*E. vignae*	*V. radiata*	MT628282	MT628017	[20]
*E. vignae*	*V. radiata*	MW293895	MT628018	[20]
*E. vignae*	*V. radiata*	MT628285	-	[20]
*E. vignae*	*V. radiata*	MT628286	-	[20]
*E. vignae*	*V. radiata*	MW293894	-	[20]
*Erysiphe* sp.	*P. vulgaris*	AY739109	-	[15]
*Pseudoidium kalanchoes*	*Kalanchoe blossfeldiana*	-	MK411006	[28]
*P. pedaliacearum*	*Sesamum indicum*	-	LC342967	[35]
*Salmonomyces* sp.	*Acalypha wilkesiana*	-	MT133550	[20]
*Uncultured Erysiphe*	*P. vulgaris*	KU320678	-	Unpublished
*Uncultured Oidium*	*P. vulgaris*	HQ444195	-	[12]
*Uncultured Oidium*	*P. vulgaris*	HQ444197	-	[12]
*Uncultured Oidium*	*P. vulgaris*	HQ444198	-	[12]

**Table 2 plants-11-00874-t002:** Host range tests of common bean powdery mildew.

Species	Cultivar	Infection Type	Susceptibility ^1^
*Phaseolus vulgaris*	Yingguohong	4	(+)
	Pinjinyun 5	4	(+)
	F3370	2	(+)
	F5033	2	(+)
*Vigna unguiculata*	Guijiangdou 1805	4	(+)
	Pinjiang 2013-25-44	4	(+)
	Zhongjiang 1	3	(+)
	Jijiang 3	1	(+)
*Vigna radiata*	Jinlvdou 9	4	(+)
	Jilv 7	4	(+)
	Jilv 0816	4	(+)
	Pinlv 2014-129	4	(+)
*Vigna angularis*	Jihong 13	0	(−)
	Baihong 12	0	(−)
	Baohong 201206-5	0	(−)
	Liaohong 12814	0	(−)
*Lablab purpureus*	Jiaoda 48	3,4	(−)
	Jiaodayanhongbian	0,3,4	(−)
*Phaseolus multiflorus*	18E07	0,3,4	(−)
*Vicia faba*	Qinghai 13	0	(−)
	Yundou 1183	0	(−)
*Pisum sativum*	Zhongqing 1	0	(−)
	Longwan 1	0	(−)
*Lens culinaris*	Bendixiaobiandou	0	(−)
	Yingguozhonglv	0	(−)
*Cicer arietinum*	Xinying 1	0	(−)
	Xinying 2	0	(−)
*Glycine max*	Williams	0	(−)
	Huachun 18	0	(−)
*Cucurbita moschata*	unknown	0	(−)
*Cucumis sativus*	unknown	0	(−)

^1^ “(+)/(−)” indicated the isolate CBPW1 was pathogenic or nonpathogenic to this cultivar.

**Table 3 plants-11-00874-t003:** Identification of resistance to powdery mildew on common bean cultivars.

Cultivar	Breeding Unit	Infection Type	Reaction
LiBY-5	Bijie Academy of Agricultural Sciences	1	HR
Keyun 3	Keshan Branch of Heilongjiang Academy of Agricultural Sciences	1	HR
Long 15-1909	Institute of Crop Germplasm, Heilongjiang Academy of Agricultural Sciences	1	HR
Long 17-4167	Institute of Crop Germplasm, Heilongjiang Academy of Agricultural Sciences	1	HR
Longyundou 4	Institute of Crop Germplasm, Heilongjiang Academy of Agricultural Sciences	1	HR
Longyundou 10	Institute of Crop Germplasm, Heilongjiang Academy of Agricultural Sciences	1	HR
Longyundou 18	Institute of Crop Germplasm, Heilongjiang Academy of Agricultural Sciences	1	HR
Pinyun 2	Institute of Crop Germplasm, Heilongjiang Academy of Agricultural Sciences	2	R
Longyundou 14	Institute of Crop Germplasm, Heilongjiang Academy of Agricultural Sciences	2	R
F5033	Institute of Crop Sciences, Chinese Academy of Agricultural Sciences	2	R
F3370	Institute of Crop Sciences, Chinese Academy of Agricultural Sciences	2	R
Biyun 19-1	Bijie Academy of Agricultural Sciences	3	S
Long 15-1694	Institute of Crop Germplasm, Heilongjiang Academy of Agricultural Sciences	3	S
Long 15-1554	Institute of Crop Germplasm, Heilongjiang Academy of Agricultural Sciences	3	S
Long 15-1898	Institute of Crop Germplasm, Heilongjiang Academy of Agricultural Sciences	3	S
Longyundou 16	Institute of Crop Germplasm, Heilongjiang Academy of Agricultural Sciences	3	S
Longyundou 20	Institute of Crop Germplasm, Heilongjiang Academy of Agricultural Sciences	3	S
ZYD19-02	Institute of Crop Sciences, Chinese Academy of Agricultural Sciences	3	S
Baiyun 3	Baicheng Academy of Agricultural Sciences	4	HS
LiBY-1	Bijie Academy of Agricultural Sciences	4	HS
LiBY-2	Bijie Academy of Agricultural Sciences	4	HS
LiBY-3	Bijie Academy of Agricultural Sciences	4	HS
LiBY-7	Bijie Academy of Agricultural Sciences	4	HS
LiBY-8	Bijie Academy of Agricultural Sciences	4	HS
LiBY-10	Bijie Academy of Agricultural Sciences	4	HS
LiBY-11	Bijie Academy of Agricultural Sciences	4	HS
LiBY-12	Bijie Academy of Agricultural Sciences	4	HS
LiBY-13	Bijie Academy of Agricultural Sciences	4	HS
LiBY-14	Bijie Academy of Agricultural Sciences	4	HS
BY-6	Bijie Academy of Agricultural Sciences	4	HS
BY-7	Bijie Academy of Agricultural Sciences	4	HS
BY-8	Bijie Academy of Agricultural Sciences	4	HS
BY-9	Bijie Academy of Agricultural Sciences	4	HS
Biyun 19-2	Bijie Academy of Agricultural Sciences	4	HS
Qianyundou 1	Bijie Academy of Agricultural Sciences	4	HS
Long 15-1858	Institute of Crop Germplasm, Heilongjiang Academy of Agricultural Sciences	4	HS
Long 15-1604	Institute of Crop Germplasm, Heilongjiang Academy of Agricultural Sciences	4	HS
Long 15-1607	Institute of Crop Germplasm, Heilongjiang Academy of Agricultural Sciences	4	HS
Long 16-3545	Institute of Crop Germplasm, Heilongjiang Academy of Agricultural Sciences	4	HS
Long 15-3580	Institute of Crop Germplasm, Heilongjiang Academy of Agricultural Sciences	4	HS
Longyundou 5	Institute of Crop Germplasm, Heilongjiang Academy of Agricultural Sciences	4	HS
Longyundou 17	Institute of Crop Germplasm, Heilongjiang Academy of Agricultural Sciences	4	HS
Longyundou 19	Institute of Crop Germplasm, Heilongjiang Academy of Agricultural Sciences	4	HS
Longyundou 21	Institute of Crop Germplasm, Heilongjiang Academy of Agricultural Sciences	4	HS
Xiaobailian	Institute of Crop Germplasm, Heilongjiang Academy of Agricultural Sciences	4	HS
Yidianhei	Institute of Crop Germplasm, Heilongjiang Academy of Agricultural Sciences	4	HS
Pinjinyun 5	Center for Agricultural Resource Research, Shanxi Agricultural University	4	HS
Xinyun 8	Institute of Food Crops, Xinjiang Academy of Agricultural Sciences	4	HS
ZYD19-02	Institute of Crop Sciences, Chinese Academy of Agricultural Sciences	4	HS
LiBY-4	Bijie Academy of Agricultural Sciences	0, 4	IM, HS
LiBY-6	Bijie Academy of Agricultural Sciences	0, 4	IM, HS
LiBY-9	Bijie Academy of Agricultural Sciences	0, 4	IM, HS
ZYD19-1	Institute of Crop Sciences, Chinese Academy of Agricultural Sciences	0, 4	IM, HS
Long 16-3263	Institute of Crop Germplasm, Heilongjiang Academy of Agricultural Sciences	1, 3	HR, S

## Data Availability

The data presented in this study are available on request from the corresponding author.

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
