# Peer review of "Identification of Causal Agent Inciting Powdery Mildew on Common Bean and Screening of Resistance Cultivars"

_plants, 2022, doi:10.3390/plants11070874_

Round 1
Reviewer 1 Report
The manuscript by Dent et al., identified using morphological and genomic analysis the causal agent of powdery mildew on common bean as well screening to identify resistant cultivars.
I consider that information and strategy is well described and performed, but still is very descriptive fro this Journal. Morphological and genomic characterization of the pathogen is clear and well supported, as well as the host range test. However, concerning the identification of the resistant bean cultivars, I consider that the information is preliminar. Authors did not explore if the canonical defense mechanisms are similar among the resistant cultivars. I consider that for a journal with the impact as Plants this information should be included.
Minor comments.
- Line 34, "improve fertility" should be more clearly described (this by N assimilation due to symbiosis?).
- Line 105. "The genomic DNA of isolated CBPM1 was amplified", the genomic DNA was used to amplified a specific gene. Please indicate the name of the gene.
- Line 253. "narrow genetic background". Is never explained why, please include the information of the references that support this sentence.
Author Response
Response to Reviewer 1 Comments
Thank you very much for providing much valuable and constructive comments and suggestions. We felt all of the comments provided were highly helpful for revising and improving this manuscript. We have studied the comments and suggestions carefully and revised our manuscript in detail according to the comments and suggestions that you have provided. The revisions were marked in blue color for Reviewer 1 in this revised manuscript. Meanwhile, I changed the fonts in Figures 3 and 4 to match the format.
Minor comments.
Point 1: Line 34, "improve fertility" should be more clearly described (this by N assimilation due to symbiosis?).
Response 1: Thanks very much for your careful and professional suggestions. The common bean can improve soil fertility by N assimilation due to symbiosis. (in blue)
Point 2: Line 105, "The genomic DNA of isolated CBPM1 was amplified", the genomic DNA was used to amplified a specific gene. Please indicate the name of the gene.
Response 2: We are sorry for our carelessness. We changed “genomic DNA” to “ITS region”. Thanks very much for your careful comments. (in blue)
Point 3: Line 253, "narrow genetic background". Is never explained why, please include the information of the references that support this sentence.
Response 3: Thanks for your good comment. The 54 common bean cultivars, mostly from the two breeding units, were selected from a small number of parents. We added the original information of each cultivars in Table 3 to explain the cultivars clearly. (in blue)

Reviewer 2 Report
Deng et al. performed this study on “Identification of Causal Agent Inciting Powdery Mildew on Common Bean and Screening of Resistance Cultivars” to identify the pathogen causing powdery mildew in kidney beans. Although major focus was kidney beans, this study has majorly focused on determining the host range while confirming the isolate CBPM1 as causative agent. I hope the following comments will help authors in improving the comprehensiveness, scientific soundness and clarity of the manuscript.
Major comments:
In the introduction, the main focus was powdery mildew in Phaseolus vulgaris however the experiments were designed to test the host-pathogen interaction in multiple leguminous crops. It is difficult to connect these two aspects.
What was the basis of selection of 54 cultivars for resistance study? Did you have aprior information for this collection on this study?
Statistical analyses is completely missing from the study. How did you include replicates and repetitions and what analyses were performed to call them of particular disease score?
Minor comments:
Line 32: Explain: direct human consumption
Line 51: most closely genetic relationship?
Line 138: phylogenetiv analyses
Line 265: Explain “A single spot was …”
Line 268: “powdery mildew” did you mean fungal spores/conidia?
Line 293: Mention the sample/project number and provide NCBI references for the deposited sequences
Author Response
Response to Reviewer 2 Comments
Thank you very much for providing much valuable and constructive comments and suggestions. We felt all of the comments provided were highly helpful for revising and improving this manuscript. We have studied the comments and suggestions carefully and revised our manuscript in detail according to the comments and suggestions that you have provided. The revisions were marked in red for Reviewer 2 in this revised manuscript.
Major comments:
Point 1: In the introduction, the main focus was powdery mildew in Phaseolus vulgaris however the experiments were designed to test the host-pathogen interaction in multiple leguminous crops. It is difficult to connect these two aspects.
Response 1: Thanks for your good comment. The agent of powdery mildew on common bean was generally considered to be Erysiphe polygoni, while recent research by Kelly et al [20] showed that E. vignae was an agent of common bean powdery mildew. E. vignae could also infect mungbean, black gram. One objective of this study is to identify the pathogen cuasing powdery midew on common bean in China. Test of host range is necessary for identication of plant pathogen.
Point 2: What was the basis of selection of 54 cultivars for resistance study? Did you have a prior information for this collection on this study?
Response 2: Thanks very much for your careful and professional suggestions. The information of the cultivars were added in Table 3 (In blue). These 54 common bean cultivars were collected from the main common bean breeding units in China.
Point 3: Statistical analyses is completely missing from the study. How did you include replicates and repetitions and what analyses were performed to call them of particular disease score?
Response 3: Thanks for your good comment. According to most of previous studies, evaluation for resistance to common powdery mildew on common bean was based on powdery mildew infection types (five score: 0 to 4), which was not necessary to use statistical analysis. In this study, pathogenicity and host range tests were repeated twice, and the cultivars with the infection types of 0-2 were identified twice.
Minor comments:
Line 32: Explain: direct human consumption
Response 4: The ‘direct human consumption’ means common bean as main food can be eatan directly. Many references described common bean as the followings:
- ‘The common bean (Phaseolus vulgaris) is the most important grain legume for direct human consumption, being especially important in eastern Africa and in Latin America.’ (Beebe et al. 2000)
- Common bean (Phaseolus vulgaris L) is the most important food legume for direct human consumption, provides significant quantities of protein and energy, and is a source of vitamins and minerals including Fe and Zn. (Basavaraja et al. 2021)
- Common bean (Phaseolus vulgaris) is the most important legume for human consumption worldwide and an important source of vegetable protein, minerals, antioxidants, and bioactive compounds. (Karavidas et al. 2022)
References
- Beebe, S.; Gonzalez, A.V.; Rengifo, J. Research on Trace Minerals in the Common Bean. Food Nutr. Bull. 2000, 21, 387-391, doi:https://doi.org/10.1177/156482650002100408.
- Basavaraja T.; J. S.N.S.; Chandora R.; Singh M.; Singh N.P. (2021) Breeding for Enhanced Nutrition in Common Bean. In: Breeding for Enhanced Nutrition and Bio-Active Compounds in Food Legumes; Gupta D.S.; Gupta S.; Kumar J.; Eds Springer: Cham, Switzerland, 2021; pp. 181-209, doi:https://doi.org/10.1007/978-3-030-59215-8_8.
- Karavidas, I.; Ntatsi, G.; Vougeleka, V.; Karkanis, A.; Ntanasi, T.; Saitanis, C.; Agathokleous, E.; Ropokis, A.; Sabatino, L.; Tran, F.; Iannetta, P.P.M.; Savvas, D. Agronomic Practices to Increase the Yield and Quality of Common Bean (Phaseolus vulgaris): A Systematic Review. Agronomy2022, 12, 271, doi:https://doi.org/10.3390/agronomy12020271.
Line 51: most closely genetic relationship?
Response 5: Thanks for your good comment. The phylogenetic tree, constructed by Almeida et al. [15], based on ITS sequences showed that Erysiphe sp. isolate EB2004 had the most closely genetic relationship with the soybean powdery mildew E. diffusa.
Line 138: phylogenetiv analyses
Response 6: Thanks for your good comment. The Phylogenetic Analyses was in the Line 105-139. Line 140-162 were the Pathogenicity and Host Range Tests.
Line 265: Explain “A single spot was …”
Response 7: Thanks for your good comment. A single spot was a disease spot, which was produced by a single conidium. We changed “A single spot was …” to ‘’Conidia producing on a single diseased spot’’. (in red)
Line 268: “powdery mildew” did you mean fungal spores/conidia?
Response 8: Thanks for your good comment. “powdery mildew” means conidia/spores, and contains conidiophore, hypha, and so on.
Line 293: Mention the sample/project number and provide NCBI references for the deposited sequences
Response 9: Thanks. The information of related powdery mildew strains sequences were all in the Table 1.

Round 2
Reviewer 2 Report
I would like to thank Deng et al. for making suggested changes in the manuscript. I do not have any other comments.